# Mapping Trends in Drowning Research: A Bibliometric Analysis 1995–2020

**DOI:** 10.3390/ijerph18084234

**Published:** 2021-04-16

**Authors:** Justin-Paul Scarr, Jagnoor Jagnoor

**Affiliations:** 1Royal Life Saving Society—Australia, Broadway, NSW 2042, Australia; 2The George Institute for Global Health, University of New South Wales, Newton, NSW 2042, Australia; jjagnoor@georgeinstitute.org.au; 3Jasola District Centre, The George Institute for Global Health—India, University of New South Wales, New Delhi 110025, India

**Keywords:** drowning, drowning prevention, injury, bibliometric analysis, publication trends

## Abstract

Drowning is public health issue requiring global, national and community responses. The multisectoral nature of drowning prevention reinforces the need for multidisciplinary research, which can play a key role in identifying patterns, factors and interventions and contributes to evidence-informed prevention. This study presents a biometric analysis of drowning research published in 1995–2020 and identifies temporal trends in research themes, journals, countries and authorship to assist in the planning of future research. This study identified 935 studies, representing authors from 80 countries. Publications grew 103-fold, and 41.2% (*n* = 385) were published since 2014. The top 20 journals are all injury prevention, public health, or medical journals. The top 5 accounted for 24.5% (*n* = 229) of total publications (TP). Research from the United States (TP = 313, 25.0%) and Australia (TP = 192, 15.3%) dominates the field. Growth is highest in low–middle-income countries (LMICs) including China (TP = 54, 4.3%, 32-fold), India (TP = 30, 2.4%, 17-fold) and Bangladesh (TP = 47, 3.7%, 7-fold). The study identifies significant growth in epidemiologic studies reporting burden and risk factors. Research in LMICs is increasing but lags relative to the burden. The role of multilateral and nongovernment organisations in evidence generation is evident and needs investigation, as do gaps in evidence for interventions and partnerships to progress the drowning prevention field.

## 1. Introduction

Drowning has been described as ‘a highly preventable public health challenge that has never been targeted by a global strategic prevention effort’ [1]. Drowning prevention is multisectoral in nature, intersecting with key international themes including climate change; migration; child and adolescent health; water, sanitation and hygiene; and disaster. Multidisciplinary research is critical for identifying the patterns of drowning, investigating prevention measures and engaging with relevant sectors.

Global drowning is estimated to have declined by 44.5% between 1990 (*n* = 531,956) and 2020 (*n* = 295,210) [2], although the precision of health surveillance in capturing drowning mortality and morbidity is debated [3,4]. Studies using verbal autopsy in Bangladesh [5] and India [6] have highlighted a previously hidden burden among children in low–middle-income countries (LMICs), prompting increased interest by governments, multilateral organisations such as the World Health Organization (WHO) and United Nations Children’s Fund (UNICEF), nongovernment organisations and donors.

Significant advances have included the development of a new drowning definition in 2005 [7]. This was followed by increased interest in the drowning burden in LMICs, principally from across Asia, as evidenced in the WHO World Report on Child Injury Prevention [8]. The WHO’s first Global Report on Drowning: Preventing a Leading Killer in 2014 provided a catalyst for increased advocacy, research and programs [1]. As drowning prevention gains space in global health advocacy, research agendas will play a more critical role. Understanding the depth of research throughout this period may provide a basis for conceptualising future research activities as well as the development of the field.

Bibliometric analysis is used to quantitatively assess academic research activity, and to analyse and describe trends in the development of a field, and can be an effective tool for policymakers. It has been used in recent times to analyse research in fields including global health [9], intimate partner violence [10], road traffic injury [11,12] and disaster risk reduction [13]. Bibliometric findings provide a simple, objective and verifiable measure of research performance. In addition, the research contributions of different countries can be identified and used by politicians, media and evaluation agencies when assessing scientific activity and commitment to an issue on national and global levels. In recent years, bibliometric analysis has been used to inform research policy and management decisions [14].

To our knowledge, bibliometric analysis has not previously been applied to drowning. This bibliometric analysis aims to map trends in drowning research published between 1995 and 2020. The objective is to identify trends in research, journals, countries, authorship, co-authorship and key themes.

## 2. Methods

### 2.1. Search Strategy

Electronic searches for the term “drowning” in keyword, title and abstracts were conducted in PubMed, The Cochrane Library, Web of Science and Embase. This study used search results for literature published between 2005 and 2020 as part of an unpublished scoping review. To map trends over time, a second search using the same search strings, databases and inclusion/exclusion criteria for literature published between 1995 and 2004 was conducted. The process is outlined in Figure 1.

### 2.2. Inclusion and Exclusion

Studies published in English in peer-reviewed journals were included if they were systematic reviews, randomised and non- randomised controlled trials, cohort studies, case-control studies, cross-sectional studies, retrospective analysis, observational studies and qualitative studies.

Forensic studies; case reports; animal studies; studies with a singular focus on water quality/geography; studies with a singular focus on intentional drowning; comments, editorials, or letters; conference or congress papers (if abstract only); and journal news publications were excluded. Studies were selected using title screening and duplicate removal in Endnote, followed by title and abstract blinded screening by both the authors in Rayyan, a web application used for screening and coding studies in a review. Any conflicts were resolved by both authors. References were imported to Endnote; duplicates were removed, and full-text updates were completed.

### 2.3. Matching Records to the Scopus Database

VOSviewer, a computer program used for constructing bibliometric maps [15], was tested by importing data from Endnote, Web of Science and Scopus. Scopus was chosen due to the completeness of citation data. Search strings were created by extracting Digital Object Identifier (DOI) and PubMed Identifier (PMID) from Endnote. Titles were used for records without DOI or PMID. The search strings were applied to Scopus advanced search, and the results were combined to yield a list of matched publications. Data were extracted in Excel.

### 2.4. Periods for Analysis

Four periods for analysis were identified: 1995–2004, 2005–2014, 2015–2020 and 1995–2020. The first period (1995–2004) reflects the decade prior to the publication of the new medical definition of drowning. The second period (2005–2014) reflects the decade prior to the publication of the WHO Global Report on Drowning. The third period (2015–2020) reflects six years since the WHO Report. The total study period presents 26 years of research. These four periods provide a basis for temporal analysis of trends.

### 2.5. VOSviewer, Tables and Maps

VOS viewer (v1.6.16) was used to collate data and create visualisations. Text files for the periods 1995–2020, 1995–2004, 2000–2014 and 2015–2020 were applied using the biometric (Scopus) import function of VOS viewer. Thesaurus files were used to standardise duplicates and correct spellings. Total publications (TP), total citations (TC) and link strength (LS) were tabulated for countries, journals, author keywords and authors. Link strength is calculated as total publications between two authors or author keywords. Visualization maps were created for keywords and authors. These are represented by circles, where the diameter is determined by the number of occurrences. The distance between two circles represents the closeness of any association. Links between circles are represented by lines.

Linked authors or keywords are automatically clustered and assigned colours. Default settings were used in all cases except in co-author visualisation, where settings were adjusted to show authors with three or more publications. This was necessary given the lower ratio of papers per author in the 1995–2004 and 2005–2014 periods.

## 3. Results

### 3.1. Search and Scopus Matching Results

A total of 777 studies were extracted from 3346 search records identified for the period 2005–2020. This search was supplemented by a second search for studies published between the period 1995–2004, which produced 1294 studies. Duplicate and title screening reduced this to 340 potentially relevant studies. Titles and abstracts were independently screened, resulting in 186 studies for 1995–2004. The results for both periods (1995–2004, 186; 2005–2020, 777) were combined and five duplicates were removed, resulting in 958 studies being identified for biometric analysis.

Of the 958 studies, 935 (97.6%) were matched to records in the Scopus database. DOI search yielded 823 of 854 records (96.4%) and PMID identified 814 of 886 (91.9%) records. A title search for publications without a DOI or PMID ID yielded an additional 11 records. The leading Scopus subject classifications were Medicine (61.5%), Social Sciences (11.3%), Nursing (4.6%) and Environmental Sciences (4.5%) (data not shown).

### 3.2. Growth in Publications

The study identified 935 publications between 1995 and 2020. Figure 2 shows that publications grew 103-fold over the 26-year period, from nine in 1995 to 80 published in 2020. The period 2015–2020 (six years) accounts for 41.2% (*n* = 385) of all the publications since 1995.

### 3.3. Journals

Figure 2 presents the top 20 journals by publication number by year within the study period. The top five journals accounted for 24.5% (TP = 233) of publications. The top five journals by publication were *Injury Prevention* (TP = 89, 9.5%), *International Journal of Injury Control and Safety Promotion* (TP = 52, 5.6%), *BMC Public Health* (TP = 35, 3.7%), *Accident Analysis Prevention* (TP = 27, 2.9%) and *Resuscitation* (TP = 26, 2.8%). The top 20 journals are all injury prevention, public health, or medical journals. The *Journal of Coastal Research* (rank = 33, TP = 6, 0.6%) and *Natural Hazards* (rank = 41, TP = 5, 0.5%) are two nonmedical or public health journals in the top 50, and 322 individual journals were identified (data not shown).

### 3.4. Countries

Analysis of publications by country reports on the relative strength and prominence of research in a country relative to the field. The study identified 80 countries in author affiliation data, which grew from 37 in 1995–2004 to 79 in 2015–2020. Figure 3 shows that research is concentrated in two countries: the United States (TP = 313, 25.0%) and Australia (TP = 192, 15.3%). China (TP = 54, 4.3%; China was reclassified upper-middle income in 2015 by The World Bank), Bangladesh (TP = 47, 3.7%) and India (TP = 30, 2.4%) are the only low–middle-income countries to appear in the top 10. Growth rates are highest in China (32-fold), India (17-fold) and Bangladesh (7-fold).

### 3.5. Most Commonly Cited Publications

The top five most cited publications were Mokdad (2016), Jonkman (2005), Agran (2003), Flores (2010) and Jonkman (2009). Research topics included global burden of disease injuries in young people, flood disaster deaths, pediatric injuries in children 0–3 years, racial disparities in child health and mortality after hurricane Katrina. Many of these contexts may be considered distal to drowning. Linked citations provide an indication of related studies in a similar field and may be an indicator of studies proximal to the drowning field. The highest linked citation scores in Table 1 include Quan (2003, LC = 40), van Beeck (2005, LC = 29), Idris (2003, LC = 22), Brenner (2001, LC = 22) and Szpilman (2012, LC = 21).

### 3.6. Author Keywords

Author keywords reflect the intended subject and audience for the publication as selected by the authors. The study identified 1412 unique author keywords. The top five were drowning (*n* = 263, 18.6%), children (*n* = 76, 5.4%), injury (*n* = 75, 5.3%), mortality (*n* = 67, 4.7%) and epidemiology (*n* = 57, 4.0%). Those which may be considered unique to drowning included drowning prevention (*n* = 33, 2.3%), water safety (*n* = 23, 1.6%), lifeguards (*n* = 20, 1.4%), swimming pools (*n* = 14, 0.9%) and swimming (*n* = 13, 0.0%). Figure 4 shows the top author keywords with five or more instances (*n* = 63) and organised into seven clusters by VOS viewer. Leading themes (clusters) include Medical (*n* = 344, 24.4%); Child Injury and Drowning (*n* = 245, 17.4%); Drowning and Injury Prevention (*n* = 169, 12.0%); Mortality, Autopsy (*n* = 133, 9.4%); and Resuscitation, Rescue (*n* = 104, 7.4%).

### 3.7. Authors

Publication numbers and citations can be a measure of author productivity, as well as the depth of the field. The study identified 935 papers written by 2840 authors. Figure 5 shows the top 15 authors by number of publications. Leading authors included Franklin R.C. (TP = 48), Peden A.E. (TP = 36), Rahman A. (TP = 29), Quan L. (TP = 28) and Rahman A.K.M.F. (TP = 25). Quan L. (TC = 870), Bierens J.J.L.M. (TC = 832), Szpilman D. (TC=798), Franklin R.C. (TC = 653) and Smith G.S (TC = 523) where included when total citations were considered. Rahman A.K.M.F (TC = 480), Arifeen S.E. (TC = 382) and Rahman A. (TC = 345) are the only LMIC-based authors to appear in the top 10.

### 3.8. Co-Author Clusters

The existence of links between authors across the study periods is shown in Figure 6. Author clusters for the period 1995–2005 show research centred around Smith G.S., Langely J.D. and Quan L. in the United States; Arifeen S.E. in Bangladesh; and Mitchell R.J. in Australia. Throughout 2005–2014, the Bangladesh cluster expanded to include Hyder A.A. and linked to a second cluster around Rahman A.K.M.F. The Quan cluster expanded to include Bierens J.J.L.M. and Moran K. In Australia, Brander R.W. linked with Mitchell R.J., and a separate cluster emerged around Franklin R.C. Throughout 2015–2020, clusters expanded rapidly. Research in Bangladesh added authors and research collaborations, including those centred around Rahman A., Alonge O. and Jagnoor J. The Franklin R.C. cluster in Australia expanded to include Peden A.E. and linked to Brander R.W., Hamilton K., and Watt K. Another Australian cluster linked to Crawford G. and Nimmo L. emerged. A cluster emerged in China, and another cluster emerged around Spanish- and Portuguese-speaking authors Szpilman D. and Abelairas-Gomez C. The visualisation for the period 1995–2020 shows the interconnectedness of the three periods (Figure 6).

## 4. Discussion

Growth in publications, countries, authors and co-authorships provides measures of the development of a field. The results show a 100-fold increase in drowning publications since 1995. This growth is well short of that shown in bibliometric analyses conducted in the fields of disaster resilience and climate change resilience, where publications grew 265-fold and 254-fold, respectively, between 1991–2019 [13].

The Global Burden of Disease 2017 study attributed more than 51.0% of drowning mortality to China, Bangladesh, India and Pakistan; however, only 11.1% of all drowning research publications are representative of authors from these populations [2]. Authors from the United States and Australia are responsible for 39.0% of drowning research publications, and these countries have made significant investments in capacity for drowning research, as well as systems for injury surveillance.

### 4.1. Growth in 1995–2004

Temporal analysis identified that the 1995–2004 period was characterised by research by authors from countries classified as high income (HIC) by the World Bank, as 63.0% of papers originated in the United States (43.8%), Australia (11.9%) and the United Kingdom (6.0%). Much of that research focused on national representative data [16], the epidemiology of child drowning [17,18], preventing backyard pool immersions through fencing [19], swimming and water safety lessons [20] and the role of alcohol in drowning [21].

During this period, research emerged from Bangladesh, laying a foundation for a growing focus on drowning in LMICs, including a verbal autopsy study that found drowning was the leading cause of death in children 1–4 years (18.9%) [5]. In 2000 WHO estimated that 449,000 people drowned worldwide (7.4 per 100,000 population), and 97.0% of the burden occurred in LMICs [22]. This corresponded to debate [23] and the development of a new medical definition stating that drowning ‘is the process of experiencing respiratory impairment from submersion/immersion in liquid’ in the latter part of the decade [7,23].

### 4.2. Growth in 2005–2014

Drowning research increased 100-fold in the 2005–2014 period. This growth included methodological advancements in the reporting of drowning burden in LMICs, including Bangladesh, China, India, Thailand, South Africa and Brazil. Research themes in HICs grew around boating, watercraft and lifejackets; water safety; drowning risk in migrants; and lifeguard surveillance. A key study explored the association between swimming lessons and drowning prevention [24].

During this period, a WHO and United Nations International Children’s Emergency Fund (UNICEF) report on child injury identified large gaps in evidence for interventions in LMICs [8]. A Bangladesh study outlined the effectiveness and cost-effectiveness of a program implementing swimming and water safety lessons alongside daycare in rural settings [25]. The WHO published its first Global Report on Drowning, which highlighted the need for community-based interventions, effective policy and legislation and further research [1]. The study found that 40% of literature has been published since 2014. The degree to which the report has influenced the drowning prevention field and research agendas needs further investigation.

### 4.3. Growth in 2015–2020

The period 2015–2020 saw phenomenal growth (41.8%) in drowning research publications. Countries represented grew from nine in 1995 to 79. Epidemiological studies were published mapping national burden [26], drowning in children [27,28], inland waterways [29] and scuba diving [30]. Further studies focused on rescue by lifeguards [31,32], bystanders [33] and drones [34]. This period saw advancement in donor support for LMIC research, including large-scale trials in Bangladesh, and qualitative studies anchoring drowning research to context and sustainability in Bangladesh, India and Vietnam.

Research on nonfatal drowning escalated in response to the lived experiences of families impacted by drowning, availability of hospital data and growing policy inertia as fatal child drowning was reduced in HICs such as Australia [35]. Nonfatal drowning research in LMICs found a disproportionate incidence in children [36]. The lack of robust methodologies highlighted the need for a standardised classification system for nonfatal drowning that can be used across diverse contexts. Future strengthening of health systems in LMICs could mean a potential shift of drowning burden from fatal to nonfatal outcomes of morbidity and disability.

### 4.4. Implications for Future Research

There is an increase in research mapping intersections with key public health and development issues such as disaster risk reduction [37]; transport [38]; and water, sanitation and hygiene [39]. However, these exploratory studies need to be supported with empirical data to inform changes in practice.

There is mounting recognition of the role of social, political and environmental determinants of health and a need to document interlinkages between health determinants, drowning risk and the co-benefits of drowning interventions in progressing in partnership with the United Nations Sustainable Development Agenda [40]. For example, the intersection of drowning and flooding is evident in a key study that found that drowning was the cause of 44% of flood-related deaths in Asia [41]. These studies call for investments in disaster risk reduction, including in infrastructure, warning systems and community resilience. The degree to which these measures prevent drowning needs investigation.

The trend analysis highlights challenges in the sustainability of drowning research and program implementation in LMICs. These observed relationships between funding and research output reinforce the need to leverage large-scale program investment to areas of policy setting, sectoral cohesion and the political determinants of drowning prevention. Drowning policy analysis is somewhat embedded in the HIC context with little space in peer-reviewed literature; however, there is a dearth of policy analysis in the LMIC context. Calls for multisectoral approaches and improvements in legislation and laws will facilitate the implementation of water safety policies and plans [1,42].

This study has identified key research themes through keyword, author and co-author analysis. These patterns have the potential to inform future research, although a more detailed analysis of research content and gaps in evidence is required. Author and co-authorship analysis points to the role that researchers play in identifying research gaps, mentoring other researchers and partnering to develop the field. Further analysis of academic disciplines, including mapping future methodological needs, may advance the field in partnership with sectors and stakeholders, including those not currently engaged in drowning prevention.

As is evident from this analysis, equitable research in LMICs, prevention and implementation research is growing, although it requires greater investment. Much of the research reflects partnerships with lifesaving and drowning prevention nongovernment organisations; growth in investments in academic pathways, including PhD programs on drowning; and changes in donor interest. Many growth periods correspond to key events such as the World Congress on Drowning in 2002, the World Conference on Drowning Prevention in 2011 [43] and the release of the Global Report on Drowning in 2014. The degree to which such partnerships and events contribute to research agendas and growth of the field requires further investigation.

The search may have missed publications distally linked to drowning but not explicitly containing drowning in the title, keywords or abstract. The inclusion of only peer-reviewed articles from indexed journals minimises the potential influence of prominent grey literature. This study is limited to publications in English, which is a weakness as a search in other languages will identify further publications, key themes and authors. Publication and citation numbers may be an imperfect measure of author productivity. In focusing on citations and index references as a proxy for influence, citation analysis does tend to prioritise older contributions that have had more time to accumulate citations.

## 5. Conclusions

This study investigates research on drowning published between 1995 and 2020 through biometric analysis, identifying tremendous growth in research over that period, notably from LMICs. The study reinforces temporal trends in drowning research and shows how themes have been established and built upon since 1995. Multilateral organisations, including the WHO and UNICEF, pivoted to research in child drowning in LMICs. The critical role of the 2014 WHO Global Report on Drowning is evident, as 40% of the literature has been published subsequent to it.

The study shows key research themes, authors and publications. The need for further in-depth understanding of research gaps is reinforced, particularly as it relates to addressing drowning in key populations at risk, interventions aligned to the WHO’s 10 actions to prevent drowning and issues that progress drowning advocacy and positioning in development contexts.

Nongovernment organisations that have a core mandate to prevent drowning through lifesaving, rescue, education and advocacy have played a critical role in evidence generation. Further analysis of relevant sectors and stakeholders and their roles and required partnerships to escalate policy prioritisation is needed.

The study findings may assist researchers, policymakers and advocates in planning and initiating research partnerships with those currently engaged in drowning prevention and provide information on the potential intersections with those sectors that are not.

## Figures and Tables

**Figure 1 ijerph-18-04234-f001:**
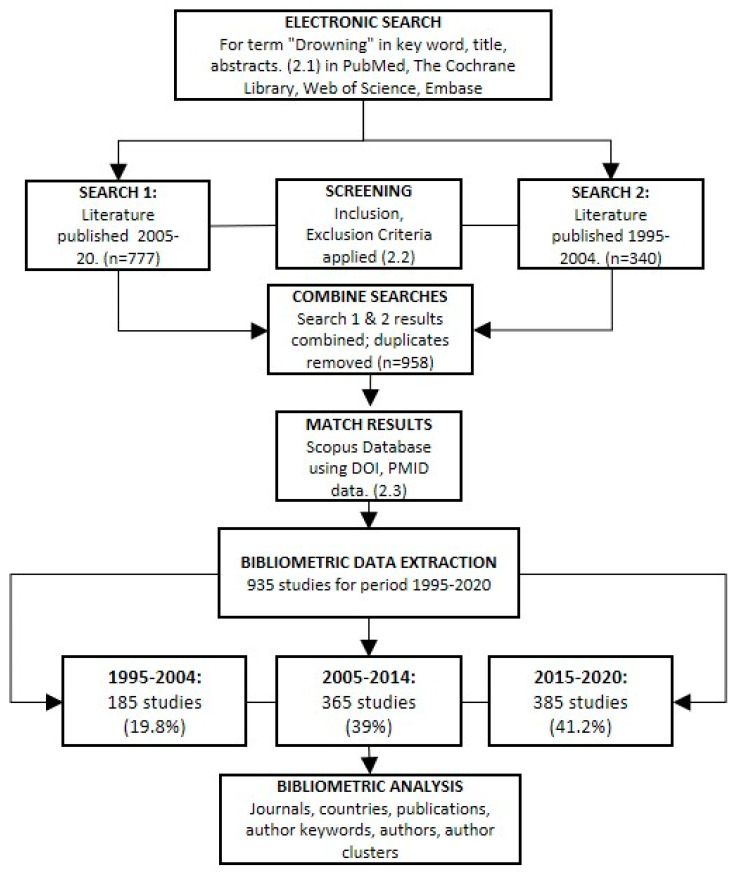
Process map for identifying literature.

**Figure 2 ijerph-18-04234-f002:**
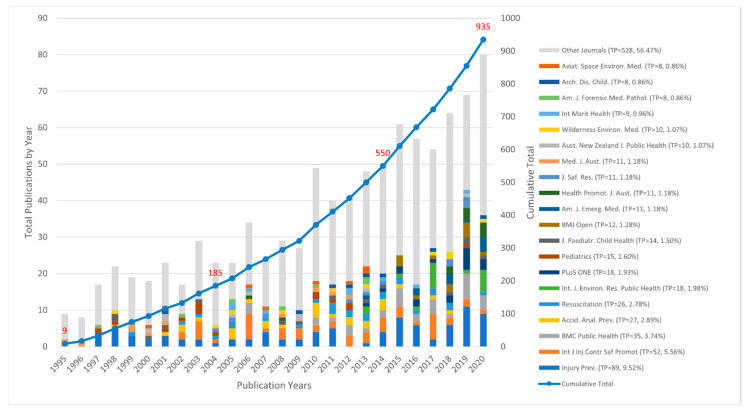
Total drowning publications by year, cumulative total and top 20 journals, 1995–2020.

**Figure 3 ijerph-18-04234-f003:**
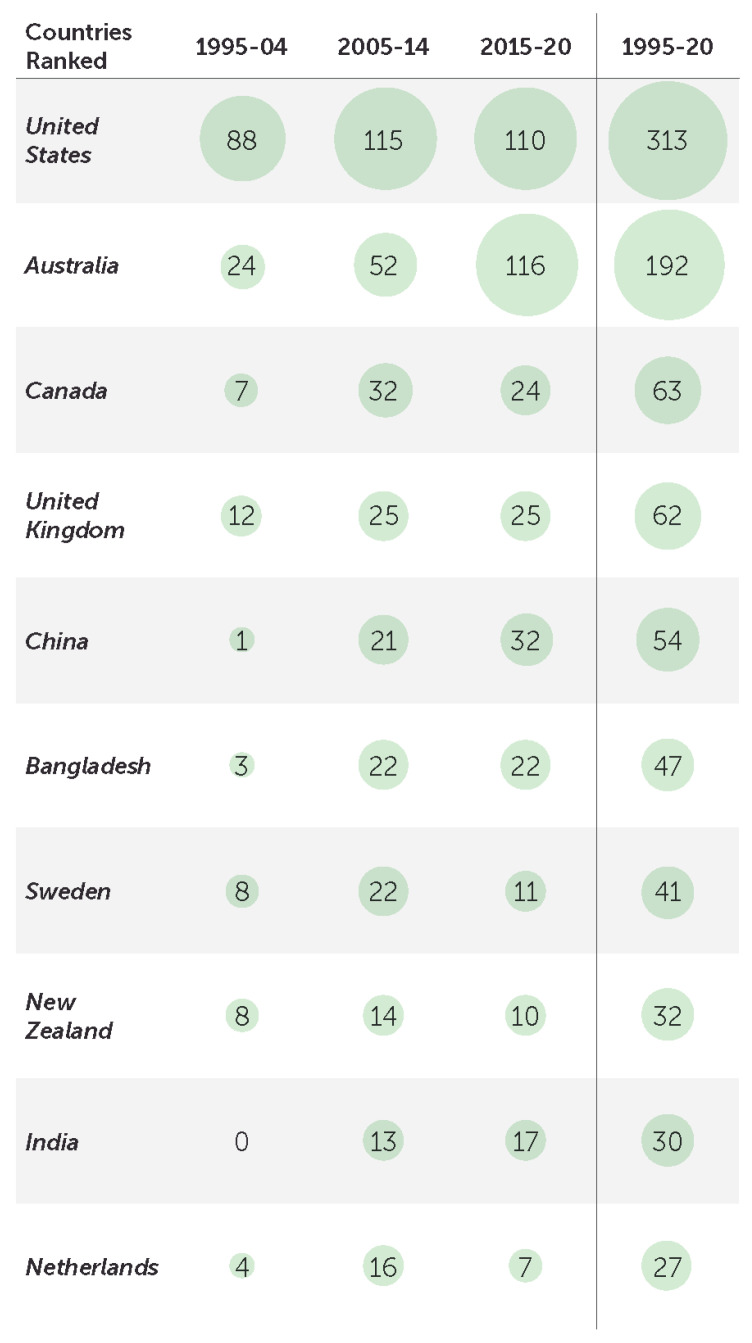
Top 10 countries represented in publications, 1995–20 and across three periods of 1995–2004, 2005–2014 and 2015–2020.

**Figure 4 ijerph-18-04234-f004:**
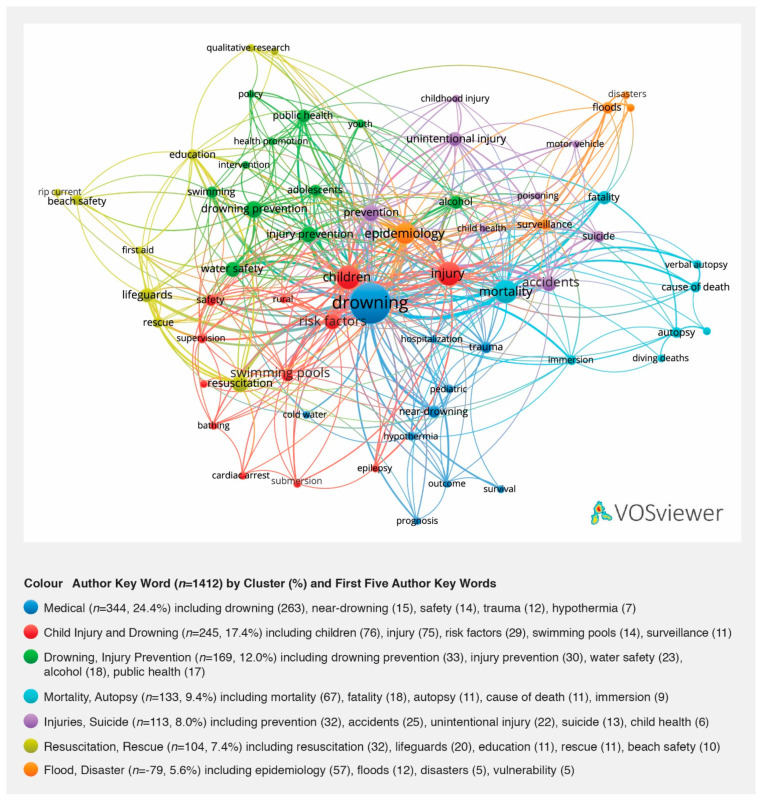
Visualisation of author keywords in seven clusters, 1995–2020.

**Figure 5 ijerph-18-04234-f005:**
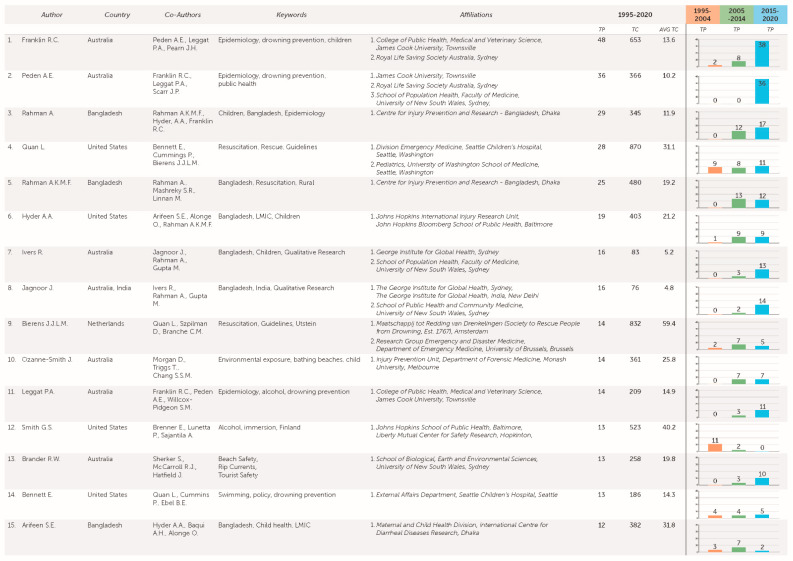
Key authors by total publications, total citations, average citations, countries, affiliations, keywords and co-authors, 1995–2020.

**Figure 6 ijerph-18-04234-f006:**
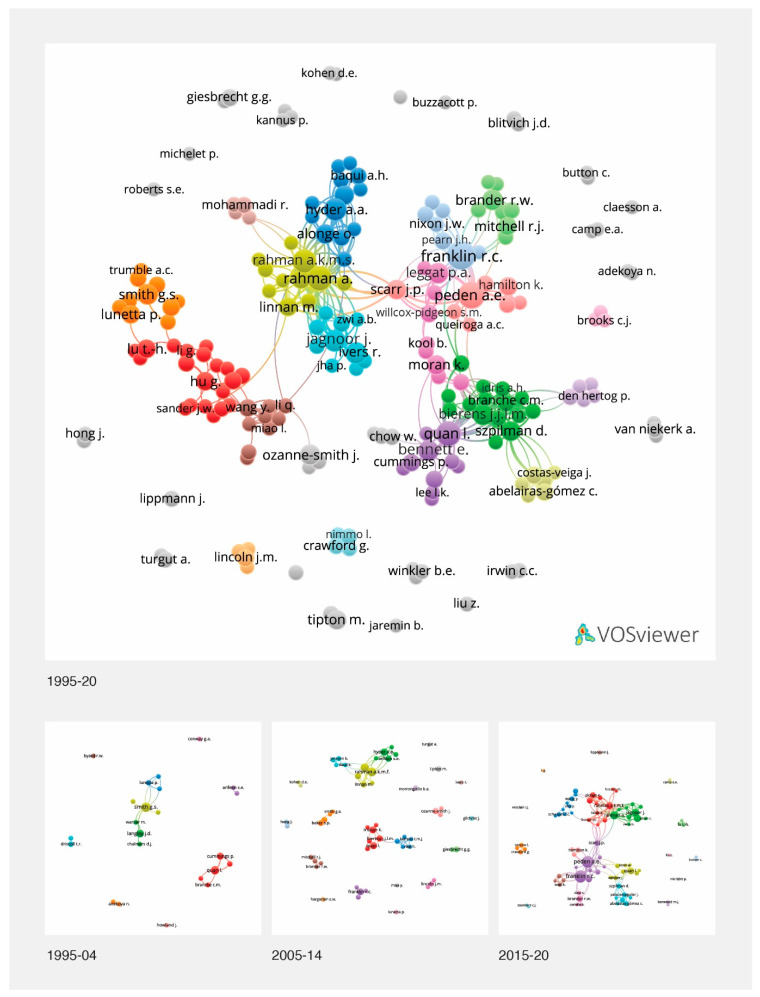
Visualisation of author clusters, 1995–2020 and across three periods of 1995–2004, 2005–2014 and 2015–2020.

**Table 1 ijerph-18-04234-t001:** Top 20 drowning publications by citation and linked citation, 1995–2020.

Author (Year)	Publication Title	Journal (Abbrev)	TC	LC
Mokdad A.H. (2016)	Global burden of diseases, injuries, and risk factors for young people’s health during 1990–2013: a systematic analysis for the Global Burden of Disease Study 2013.	*Lancet*	354	1
Jonkman S.N. (2005)	An analysis of the causes and circumstances of flood disaster deaths	*Disasters*	302	8
Agran P.F. (2003)	Rates of pediatric injuries by 3-month intervals for children 0 to 3 years of age.	*Pediatrics*	272	5
Flores G. (2010)	Racial and ethnic disparities in the health and health care of children.	*Pediatrics*	264	1
Jonkman S.N. (2009)	Loss of life caused by the flooding of New Orleans after hurricane Katrina: Analysis of the relationship between flood characteristics and mortality.	*Risk Anal*	216	3
Van Beeck E.F. (2005)	A new definition of drowning: Towards documentation and prevention of a global public health problem.	*Bull WHO*	214	29
Vyrostek S.B. (2004)	Surveillance for fatal and nonfatal injuries--United States, 2001.	*Morb Mortal Wkly*	212	1
Wang S.Y. (2008)	Injury-related fatalities in China: an under-recognised public-health problem.	*Lancet*	199	8
Szpilman D. (2012)	Drowning.	*New Engl J Med*	182	21
Brunkard J. (2008)	Hurricane Katrina deaths, Louisiana, 2005.	*Disaster Med Public Health Prep*	169	1
Eschbach K. (1999)	Death at the border.	*Int Migr Rev*	167	0
Kobusingye O. (2001)	Injury patterns in rural and urban Uganda.	*Injury Prev*	153	5
Idris A.H. (2003a)	Recommended Guidelines for Uniform Reporting of Data from Drowning the “Utstein Style”.	*Circulation*	144	22
Rimsza M.E. (2002)	Can child deaths be prevented? The Arizona Child Fatality Review Program experience.	*Pediatrics*	141	1
Thompson D.C. (2000)	Pool fencing for preventing drowning in children.	*Cochrane Database Syst Rev*	139	15
Quan L. (2003)	Characteristics of drowning by different age groups.	*Injury Prev*	134	40
Peden M.M. (2003)	The epidemiology of drowning worldwide.	*Inj Control SafPromot*	128	18
Baqui A.H. (1998)	Causes of childhood deaths in Bangladesh: Results of a nationwide verbal autopsy study.	*Bulletin WHO*	125	5
Brenner R.A. (2009)	Association between swimming lessons and drowning in childhood: A case-control study.	*Arch Pediatr Adolesc Med*	123	22
Giesbrecht G.G. (2000)	Cold stress, near drowning and accidental hypothermia: A review.	*Aviat Space Environ Med*	121	2

## Data Availability

All data relevant to the study are included in the article. Further data enquiries should be directed to the corresponding author.

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
