# Peer review of "Mapping Trends in Drowning Research: A Bibliometric Analysis 1995–2020"

_ijerph, 2021, doi:10.3390/ijerph18084234_

Round 1

Reviewer 1 Report

Thank you for this good work. 

Line 37 - I had to look up the phrase multi-laterals and think about its application to the drowning issue.  I believe it means multi-national groups. I suspect that multi-lateral might be a reasonable term here but consider the clearer term multi-national groups.

Line 41 The WHO World Report on Child Injury Prevention is the title of reference 8.  

Line 56, consider 2005-2020

Line 58, consider 1995-2004

Line 60 - consider making a comment or sentence about what % of the drowning related literature is in English vs other languages.

Line 68, Rayyan is a systematic reviews web app - consider - by both authors in Rayyan Systematic Review web app.

Line 71,  VOSviewer is a software tool for constructing and visualizing biometric networks. Consider providing this clarity.

Line 79, 80 see above comments about using full year ie 2020.

Line 100, consider removing the JS and JJ then--- and consider starting this sentence with Titles and abstracts were independently screened, resulting---

Line 252, there is a spelling, letter missing - I think this may be incidence in children.

Line 300,301, In the lines 239, 240 it states that the degree to which the WHO GROD influences the field and research agenda needs further investigation. In the conclusion you state that the critical role is evident with 40% published literature since.  A better alignment of these sections statement would be worthy of consideration.  

Thank you for this Bibliometric Analysis of an emerging field.  You should have pride in this work.  This becomes a benchmark.  I hope the suggestions for your consideration are helpful. 

Reviewer 2 Report

Comments to “Mapping trends in drowning research: a bibliometric analysis 1995-20”:

- In the title you include the period, but you must write 2020, since it is another decade. By the way, it is a period of 26 years. Correct where you must
- lines 10-11: end and start with "research". Correct
- It is not necessary to dedicate a section to the limitations, in addition to being short
- The Intro section is weak. The need for this study is not adequately justified. Why should we read it? What does it contribute beyond the subsequent presentation of annual and periodic results? What is the background? Who has studied this topic in depth?
- Excessive use is made of the personal form (we, our) in the writing. The scientific article must overcome this barrier and be written in an impersonal way.
- A graph or figure that indicates the inclusion-exclusion process would help
- In which other works this methodology has been applied with success. It would be interesting to include some so that the reader feels that the research is supported
- What is the interest of this period 1995-2020? Why not another?
- It is convenient to carry out an analysis of the financing centers that allow the growth of this topic
- Line 144: (2003a ... What is "a"?
- Table 1 ... would be "short title of jounal"
- Are the "author key words" or are they the ones that define the publication?
- In fig. 3 are the 1412 words represented or just a selection?
- Fig. 4 is interesting but not visible. To correct
- What is high income country (HIC)? What literature supports it?
- What is the justification for analyzing the total period in subperiod with durations of 10, 10 and 5? What events sustain it? What happened at each of the limits?

Round 2

Reviewer 2 Report

Comments to "Mapping trends in drowning research: a bibliometric analysis 1995-2020":
The work has improved but there are still doubtful areas, such as:
- The justification for the period analyzed is not clear. Also the subdivision in 2 decades and 6 years is arbitrary. What events justify them?
- Section 2 has subsections of 1 single sentence.
- What tools have you used to detect: Duplicate and title screening?
- Order the figures in relation to the text where they are cited
- Line 30: Sustainable Development agenda. What is it?
- Line 361-363: How can the results of this study help the agents you indicate?

Author Response

Thank you for your feedback. 

Please find below our responses and adjustments to the manuscript. 

Regards

Justin Scarr
